# Folded network and structural transition in molten tin

Liang Xu[1,2], Zhigang Wang[1], Jian Chen[2], Songyi Chen[2], Wenge Yang[2], Yang Ren[3,7], Xiaobing Zuo[3], Jianrong Zeng[4,5], Qiang Wu[1✉] & Howard Sheng[6✉]

The fundamental relationships between the structure and properties of liquids are far from being well understood. For instance, the structural origins of many liquid anomalies still remain unclear, but liquid-liquid transitions (LLT) are believed to hold a key. However, experimental demonstrations of LLTs have been rather challenging. Here, we report experimental and theoretical evidence of a second-order-like LLT in molten tin, one which favors a percolating covalent bond network at high temperatures. The observed structural transition originates from the fluctuating metallic/covalent behavior of atomic bonding, and consequently a new paradigm of liquid structure emerges. The liquid structure, described in the form of a folded network, bridges two well-established structural models for disordered systems, i.e., the random packing of hard-spheres and a continuous random network, offering a large structural midground for liquids and glasses. Our findings provide an unparalleled physical picture of the atomic arrangement for a plethora of liquids, shedding light on the thermodynamic and dynamic anomalies of liquids but also entailing far-reaching implications for studying liquid polyamorphism and dynamical transitions in liquids.

[1] National Key Laboratory of Shock Wave and Detonation Physics, Institute of Fluid Physics, China Academy of Engineering Physics, 621900 Mianyang, China. [2] Center for High Pressure Science and Technology Advanced Research, 201203 Shanghai, China. [3] X-ray Science Division, Advanced Photon Source, Argonne National Laboratory, Argonne, IL 60439, USA. [4] Shanghai Synchrotron Radiation Facility, Zhangjiang Laboratory, Shanghai Advanced Research Institute, Chinese Academy of Sciences, 201204 Shanghai, China. [5] Shanghai Institute of Applied Physics, Chinese Academy of Sciences, 201800 Shanghai, China. [6] Department of Physics and Astronomy, George Mason University, Fairfax, VA 22030, USA. [7] Present address: Department of Physics, City University of Hong Kong, Kowloon, Hong Kong. ✉email: wuqianglsd@163.com; hsheng@gmu.edu

Bernal's random close packing of equal spheres[1,2] serves as an important structural model for simple liquids, based on which liquid theories have been greatly advanced[3,4]. While many single-component liquids fall within the hard-sphere (HS) paradigm[3–6], the celebrated HS model falls short in describing metallic liquids pertaining to post-transition metals typified by tin (Sn)[6]. These liquids are known to exhibit anomalous structure factors[6–9], i.e., a low-lying shoulder on the high-angle side of the first diffraction peak. Different theoretical explanations have been proposed to rationalize this anomalous structural feature, such as overlapping hard spheres[10–12], core-polarization[13], Friedel oscillation[14], core-softened interatomic interactions[7,15,16], etc. Structure-wise, a popular view is that the liquid consists of anisotropic local structures containing two or more different structural motifs[11,12]. This view, however, has been rivaled by recent first-principles simulations ascribing the structural anomaly to short bonds in the liquid[17]. Suffice it to say, a unified atomic picture of the structure of this group of liquids is still lacking. The same can be said about other anomalous liquids, such as water[18,19] and silica[20], two very important liquids that dominate our universe, about which a consensual structural model has not yet been reached. Liquid anomalies, such as density, heat capacity, compressibility anomalies, etc.[18,21–23], are commonly associated with these liquids. Dynamic crossovers have also been found to be generic to these network-forming and glass-forming liquids[20,24–26], as exemplified by the fragile-to-strong transition[27]. While the exact structural origin of such anomalies remains elusive due to subtle changes in the structure, it has long been postulated that they are connected to LLTs[28–31]. As such, the quantification of liquid phase transitions of the first- or higher-order (continuous transition) becomes highly relevant toward establishing reliable structure-properties relationships. However, experimental verification of the LLTs proves to be highly challenging, e.g., many theoretically predicted first-order LLTs occur in deeply supercooled regions where crystallization becomes a kinetically preemptive process. Consequently, unambiguous experimental demonstrations of LLTs remain exceedingly rare in the literature[21,32–34]. As far as higher-order phase transitions are concerned, except for sulphur[35] and helium[36], there have been no reports on the existence of high-order LLTs in pure metals as of yet.

In this work, we carried out state-of-the-art synchrotron X-ray scattering experiments and advanced ab initio thermodynamic analysis to quantify the transition in the liquid Sn, in hope to address the outstanding issue of how exactly the properties of anomalous liquids are affected by the structures. We further propose a generalized folded-network structure model for Sn and similar liquids. In a broad picture, the structural model fills the gap between two existing structural models for disordered materials, i.e., the HS model and Zachariasen's continuous random network (CRN)[37], offering a unified structural description for liquid systems with mixed bonding characters. Our work establishes a new paradigm for liquid properties (e.g., quasi-universality[4] and dynamic crossovers in liquids) and the structures of disordered systems with complexity, but it will also usher in new research on liquid polyamorphism[30,38], pivoting on the concept of bond-folding.

## Results and discussion

**Structural anomalies and crossover in liquid Sn.** High-precision structure factors provide key information to unravel the structures of liquids. In this work, the structural changes of liquid Sn (*l*-Sn) were monitored by temperature-resolved high-energy X-ray diffraction (XRD) conducted in a wide temperature range. The static structure factors $S(q)$ and radial distribution functions (RDF) $g(r)$ were accurately derived as a function of temperature, as shown in Fig. 1. The details of extracting structure factors from the XRD data are given in Supplementary Fig. 1 and Note 1. Meanwhile, extensive ab initio molecular dynamics (AIMD) were carried out to provide more diagnostic information on the atomic-level structures. By comparing the high-energy XRD data with ab initio MD results (Supplementary Fig. 2), we can see that the agreement between AIMD and the experiment is exceedingly good. This bi-pronged approach enables a thorough and reliable

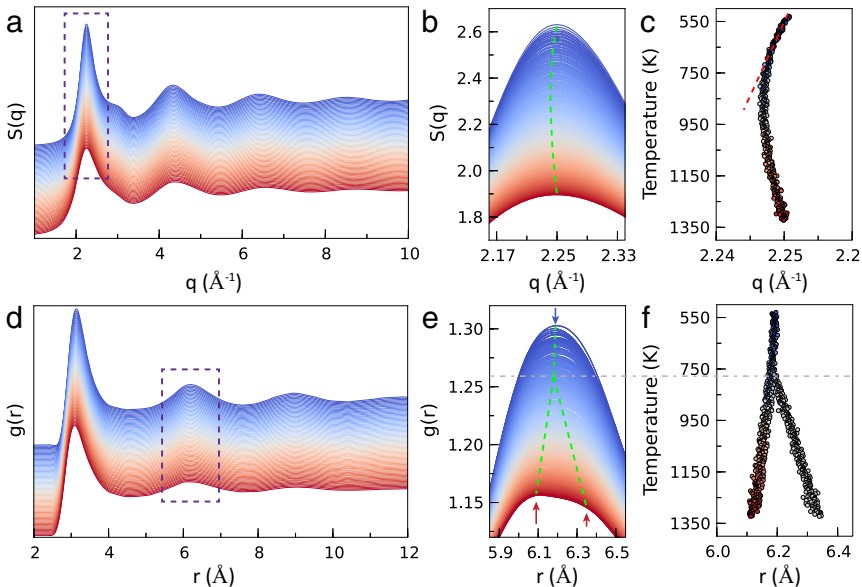

**Fig. 1 Temperature dependence of the structure of liquid Sn. a** Temperature-resolved structure factors $S(q)$ obtained from in situ synchrotron XRD from 530 K (top) to 1323 K (bottom). The structure factors were shifted vertically for visualization. **b**, **c** Magnifications of the first peak in $S(q)$ showing the temperature dependence of the peak position. The intensity of the first peak decreases with increasing temperature. An anomalous peak shift takes place at $T = 760$ K, suggestive of a structural crossover or a transition. **d** Radial distribution function $g(r)$ obtained by Fourier transformation of $S(q)$. **e** Magnifications of the second peak in $g(r)$. Splitting of the peak appears at elevated temperatures ($T > 760$ K). **f** Temperature dependence of the second peak position showing the trend of peak splitting; the main sub-peak shifts toward shorter $r$ with increasing temperature.

investigation of the structural transition of $l$-Sn. The low-lying shoulder on the $S(q)$ at around $q = 2.8\,\text{Å}^{-1}$ is consistent with previous observations[6,8], exhibiting an anomalous feature of the liquid. To highlight the evolution of the structure, we traced the shift of peak positions of $S(q)$ and $g(r)$, which are indicative of the density changes of the liquid, as shown in Fig. 1b, c. The method for finding the peak position is detailed in Supplementary Fig. 3. Upon heating, the position of the first peak of $S(q)$ continuously shifts to lower $q$ values up to 760 K, trailed by a reverse trend with increasing temperature. Such a reentrant behavior has not been reported for other liquid metals[39], hinting at a structural cross-over around 760 K.

The real-space RDF of $l$-Sn is shown in Fig. 1d at different temperatures. The peak position of the first peak shifts toward shorter $r$ monotonically upon heating (Supplementary Fig. 4c, d), exhibiting the same trend observed in other metallic liquids[39]. With increasing temperature, the principal peak broadens and becomes increasingly asymmetric with an enhanced tail in the profile of $g(r)$ for the first nearest-neighbor shell, similar to a number of other metallic or alloy melts[39]. However, a new feature emerges in the $g(r)$ of $l$-Sn at high temperatures; that is, the second peak progressively flattens and eventually splits at high temperatures, as shown in Fig. 1e, f. The slope change of the peak position at ~760 K signifies the onset of the peak split.

The splitting of the second peak on the RDF is a known feature of supercooled liquids, which becomes pronounced with decreasing temperature[40,41]. A common view is that short-to-medium-range structure order develops in supercooled liquid or glasses[40], for instance, icosahedral topological structural ordering in metallic liquids[41]. Here, the peak splitting with increasing temperature is rather counterintuitive but offers a new opportunity to interrogate the structural changes of the monoatomic liquid. The cause for the split is rationalized as a consequence of tetrahedral ordering (and bond-shortening), which will be elaborated on in a later section.

Judging from the continuous changes in $S(q)$ and $g(r)$, we rule out the possibility of first-order phase transition of $l$-Sn, as confirmed by our constant pressure AIMD simulation (see Supplementary Fig. 5). Interestingly, the AIMD density profile of liquid Sn at the constant pressure condition is best described by a piecewise linear fit, exhibiting a cusp at 840 K. The inflection of the linear fit suggests that liquid Sn undergo a structural transition in a narrow temperature range, in agreement with our XRD measurements where abnormalities in the peak position occur around the same temperature. To further ascertain the nature of this transition, we sought rigorous thermodynamic analysis as follows.

**Thermodynamics of a second-order-like transition in $l$-Sn.** Constant pressure heat capacity ($C_P$) is one of the most physical properties that furnishes information on changes in phase transitions[42]. We carried out careful specific heat measurements on molten Sn (see "Methods"). Figure 2a shows the measured molar heat capacity as a function of temperature at a constant heating rate of 4 K/min. In addition to a sharp peak indicative of melting, a weak but noticeable peak appears in the vicinity of 760 K, as illustrated in the lower inset. The overall behavior of the specific heat change, as well as the magnitude of $\Delta C_P$, has also been well captured in our AIMD simulation of $l$-Sn (see Supplementary Fig. 6). To further shed light on the changes of $C_P$ in connection to the structural transition, we carried out highly accurate calorimetric experiments with a stepwise-scanning mode (see Supplementary Fig. 7 for details), which allows sufficient time for the liquid to equilibrate. Indeed, as shown in the upper inset of Fig. 2a, a more pronounced $\lambda$-like maximum suggestive of a

second-order-like structural transition near 780 K. Moreover, a power-law fit ($C_P \sim \left| \frac{T - T_c}{T_c} \right|^{-\alpha}$) was attempted to access the critical component as in second-order phase transitions in liquid helium[36] and sulfur[35]. However, the small exponents of $\alpha_{\pm} \approx 0.04(1)$ and ~10% jump are only suggestive of a second-order-like phase transition. Because the transition is accompanied by definite $C_P$ changes, yet the diverging behavior of $C_P$ is less well-characterized, this transition may correspond to the singularity-free scenario proposed to account for liquid anomalies[43].

Another thermodynamic response function, namely, the isothermal compressibility $\kappa_T$, offers an alternative means to characterize the structural transition. In this work, we interrogated this thermodynamic quantity by conducting small-angle x-ray scattering experiments (SAXS). The details of obtaining $\kappa_T$ from SAXS data are provided in Supplementary Note 2. The derived $\kappa_T$ of $l$-Sn as a function of temperature is given in Fig. 2c. We note that a diverging behavior of $\kappa_T$ around $T_c$, as expected for a second-order phase transition was not detected with our SAXS measurements. Instead, a subtle but definite cusp of $\kappa_T$ was observed around 750 K during the heating experiment. This trend is in agreement with the changes of the specific heat (Fig. 2a) of $l$-Sn. Likewise, a noticeable change of $\kappa_T$ was captured through our comprehensive ab initio calculation (see Supplementary Note 3), as shown in Fig. 2d, where the discontinuity of $\kappa_T$ occurs around ~750 K, in consistence with the second-order-like phase transition picture of $l$-Sn.

To corroborate the observed experimental results, we further analyzed the thermodynamic quantities of $l$-Sn (more specifically, free energy, entropy, isochoric-specific heat) via a multiple histogram (MH) method[44] (see Supplementary Note 4). Figure 2e shows the probability distributions of the potential energy sampled at 18 consecutive temperatures. The isochoric heat capacity ($C_V$) can be determined by

$$k_B T^2 C_V(T) = \langle U^2 \rangle - \langle U \rangle^2 \tag{1}$$

where $k_B$ is the Boltzmann constant, $T$ is the temperature, and $U$ is the internal energy. Figure 2b shows $C_V$ as a function of temperature for $l$-Sn. Similar to experimental $C_P$, a discontinuity is found in the temperature dependence of excess specific heat $C_V^{ex}$ near the transition temperature. To gain more insight into the phase transition, we invoked the recently studied quasi-universality law of liquids[4], where the Rosenfeld–Tarazona[5] (RT) relation $C_V^{ex} \propto T^{-2/5}$ has been demonstrated to hold true for a wide of range of liquids[4]. The inset shows the linearized $C_V^{ex}$ data based on the RT relation. An abrupt change of $C_V^{ex}$ is found at the transition temperature $T \approx 800$ K, and the temperature dependence of $C_V^{ex}$ is very well described by the RT law below and above the transition temperature.

More strikingly, the excess entropy of $l$-Sn, $S^{ex}$, as obtained from the ab initio MH method, as well as the two-body excess entropy, $S_2^{ex}$, derived from the experimental RDF ($S_2^{ex} = -2\pi\rho \int_0^{\infty} \{g(r)\ln[g(r)] - [g(r) - 1]\} r^2 \mathrm{d}r$)[45] lends further support to the structural transition of the liquid. When plotted against $T^{-2/5}$, the entropic RT relation seems to be well-followed in many simple liquids[5]. In the case of $l$-Sn, the RT relation is observed piecewise linearly crossing a transition temperature, and the slope change of the linearized entropy–temperature relation (Fig. 2f) manifests as a liquid–liquid transition. At high temperatures, the reduced slope of $S_2^{ex}$ and $S^{ex}$ suggests that the liquid entropy is less than the extrapolated value from low temperatures. The reduction of the excess entropy is attributed to an increased fraction of covalent bonding at high temperatures (see below).

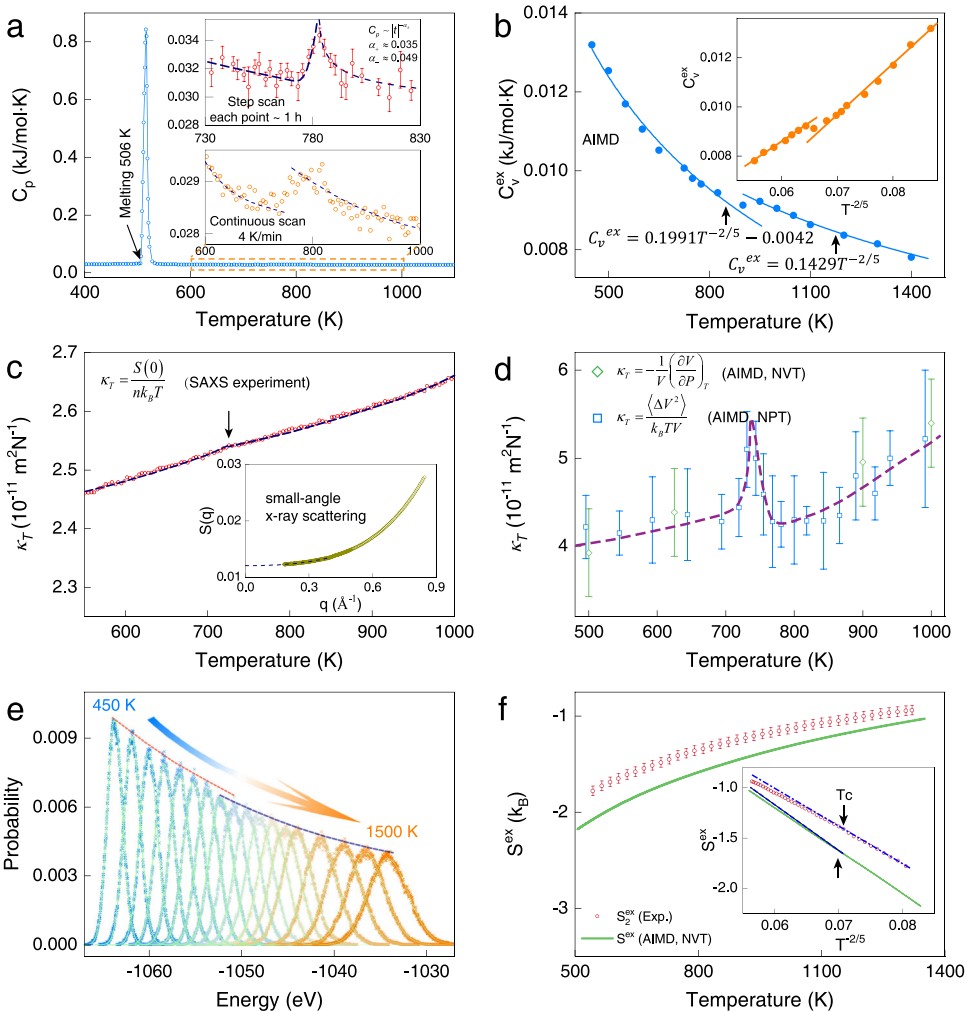

**Fig. 2 Thermodynamic properties of liquid Sn obtained from experiment and AIMD. a** Measured molar heat capacity ($C_P$) as a function of temperature at ambient pressure (see "Methods"). The lower inset shows a $\lambda$-like jump of $C_P$ at a transition temperature $T = 780$ K. The upper inset shows highly accurate $C_P$ measurements determined from the stepwise-scanning method and a critical behavior was observed near $T_c \approx 780$ K, where the exponent $\alpha_\pm = 0.035$ and 0.049, respectively. The error bars indicate the standard error of integrated heat flows taken at each temperature. **b** Isochoric-specific heat ($C_V^{ex}$) plotted as a function of temperature from AIMD. The inset shows the same data on the power scale. The solid lines are an eye-guide following the Rosenfeld–Tarazona prediction $C_V^{ex} \propto T^{-2/5}$. **c** Isothermal compressibility ($\kappa_T$) of $l$-Sn as a function of temperature derived from small-angle X-ray scattering experiments. The inset shows the optimized structure factors $S(q)$ obtained from SAXS. A quartic polynomial fit was used to obtain $S(0)$. **d** Isothermal compressibility as a function of temperature from AIMD, showing the divergence behavior of $\kappa_T$ at 750 K. The error bars correspond to the standard deviations of volume fluctuations. **e** Probability distributions of the potential energy of $l$-Sn ($V = 30$ Å$^3$/atom) sampled at different temperatures employing AIMD (symbols). The solid lines represent theoretical probability distributions resulting from the MH method. **f** Two-body excess entropy ($S_2^{ex}$) integrated from experimental RDF versus excess entropy ($S^{ex}$) obtained from the MH method of molten Sn. The error bars represent the standard errors of the data taken at each temperature. The inset shows the linearized entropy–temperature relationship following the RT relation. The second-order-like structural transition temperature from the experiment and AIMD are marked as the black arrow, where an inflection of the linearized $S^{ex}$ at $T_c = 800$ K is identified.

**Dynamic crossover associated with the LLT of $l$-Sn.** Knowledge of dynamics has strong implications for the physical state of a system. Figure 3a displays the longitudinal sound velocities ($V_L$) as a function of temperature at different pressures. At 0.75 GPa, $V_L$ varies linearly with the temperature at lower temperatures with a slope of $-0.196$ m s$^{-1}$ K$^{-1}$, and it continues with another linear temperature dependence above the transition temperature of around $T = 770$ K with a slope of $-0.454$ m s$^{-1}$ K$^{-1}$. The same trend was found at other pressures[46–48], but at higher pressures, the slope changes became more pronounced. A small, but distinct, change in the slope indicates that the physical properties (e.g., compressibility, expansion coefficient, etc.) are significantly different in the two liquid states. It is noteworthy that the same qualitative features were observed in internal friction[49] measurements at ambient pressure.

Atomic diffusion and relaxation are dynamic properties dictated by the energy landscape of the system[50]. Dynamical crossovers have been widely observed in liquids, from which distinct structural changes are arguable. The mean relaxation times calculated from AIMD as a function of the inverse temperature of $l$-Sn are plotted in Fig. 3b (details in Supplementary Note 5). The variation of the relaxation time with temperature can be generally described by an Arrhenius plot and expressed as $\tau_\alpha = \tau_0 \exp(E_a/k_B T)$, where $\tau_0$ is the characteristic relaxation time and $E_a$ is the activation energy[51]. Liquid Sn exhibits two different activated processes separated by a transition region of ~1000 K at various simulated densities. At first sight, one may surmise that the different dynamical behavior might arise from heterogeneous dynamics[52] of the liquid. Nonetheless, our four-point correlation function analysis[52] has ruled out this

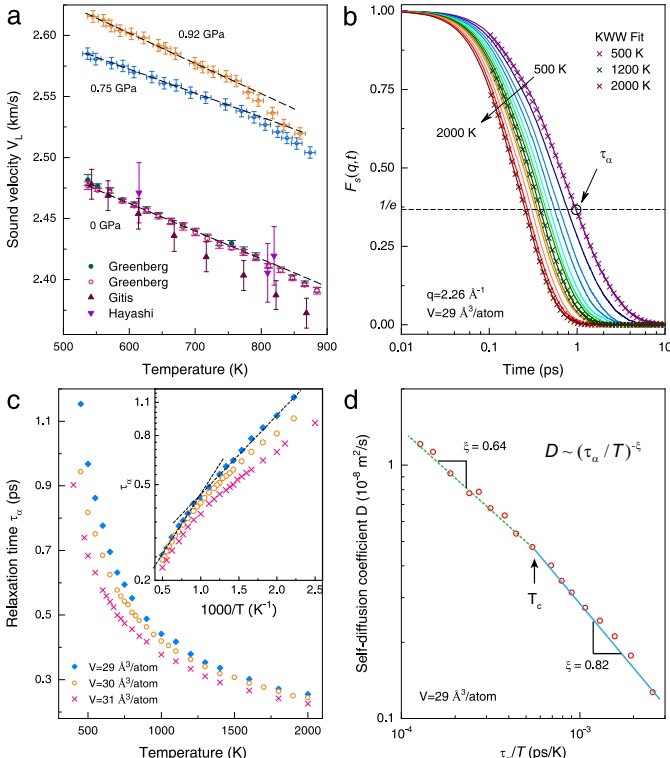

**Fig. 3 Dynamic properties of liquid Sn as a function of temperature. a** Temperature dependence of the longitudinal sound velocities at given pressures from the experiment. The data at 0 GPa were taken from Greenberg (ref. [46]), Gitis (ref. [47]), and Hayashi (ref. [48]). The error bars indicate the resolutions of data points and temperature uncertainties. **b** Intermediate scattering function $F_s(q,t)$ of liquid Sn at different temperatures obtained from AIMD. The behavior of $F_s(q,t)$ is typical of a hot liquid above its melting temperature. The $\alpha$ relaxation at late times ($t > 0.1$ ps) can be described by the Kohlrausch–Williams–Watts (KWW) relaxation function[58,59] (represented by symbols), $F_s(q, t) \sim \exp(-(t/\tau)^\beta)$, where $\tau$ is the relaxation time and $\beta$ is the stretching exponent. The relaxation time $\tau_\alpha$ is defined as the time it takes for $F_s(q,t)$ to fall to the level of $1/e$. **c** Relaxation time $\tau_\alpha$ as a function of temperature at given volumes obtained from AIMD. The temperature dependence of the relaxation time obeys the Arrhenius-type thermal activation $\tau_\alpha \sim \exp(E_a/k_B T)$, but with different activation energies $E_a$ below and above $T \approx 900$ K. Dynamic crossovers are clearly seen from the slope change of the Arrhenius fit (dashed lines in the inset). **d** Fractional SER $D \sim (\tau_\alpha/T)^{-\xi}$ of l-Sn, showing two dynamic regimes of l-Sn. At high temperatures, the dynamic behavior of l-Sn deviates more pronouncedly from the ideal SER ($\xi = 1$) with a smaller fractional exponent.

possibility. From the Arrhenius fit, the activation energy changes from 66 meV/atom at low temperatures to 93 meV/atom at high temperatures for l-Sn at V = 29 Å³/atom. The fact that liquid Sn has two dynamic regimes is consistent with our ab initio MD analysis of its atomic transport behavior invoking the Stokes-Einstein relation (SER)[53,54]. Like many other liquids[55,56], l-Sn is found to follow the fractional SER (i.e., $D \sim (\tau_\alpha/T)^{-\xi}$, where $D$ is the diffusion constant). However, as shown in Fig. 3d, the fractional exponent extracted from the fractional SER changes from 0.64 at high temperature to 0.82 at $T_c = 900$ K, manifesting two distinct dynamic regimes of l-Sn.

This high-temperature anomaly is an indication of changes in the predominant microscopic diffusion mechanism. An atomistic picture for this, however, has hitherto not been theoretically provided. We argue that the increased activation energy originates from an increased number of covalent bonds at high temperatures. On the other hand, the entropy formulation of the Adam–Gibbs theory[57] postulates that the temperature-dependent relaxation time is inversely proportional to configurational entropy, $\tau_\alpha = A\exp(B/TS_c)$, indicating that the increased relaxation time can be explained by the reduction of configurational entropy ($S_c = S_{ex} + S_c^{ideal}$, where $S_c^{ideal}$ is the entropy of an ideal gas) at high temperatures (see Fig. 2f) relative to the values extrapolated from low temperatures. The apparent dynamic and thermodynamic transition of l-Sn is deeply rooted in its structural transition[58,59].

**Core softening evidenced from the 1D radial distribution function.** To resolve the structural origin of the observed LLT in l-Sn, we set out to analyze the 1D radial distribution functions of the liquid, which were accurately obtained from the experiment and ab initio modeling.

Since the RDF $rg(r)$ satisfies a Gaussian distribution more than $g(r)$[60], we decomposed the first peak of $rg(r)$ into partial RDFs $rg_i(r)$ of individual nearest neighbor $i$. The partial RDFs are nearly symmetrical and can be satisfactorily described by Gaussian functions as shown in Supplementary Fig. 11. The decomposed $rg_i(r)$ at two selected temperatures (450 K vs. 1500 K from isochoric AIMD) are shown in Fig. 4a, from which one can see that the coordination number (CN) of liquid Sn is ~8–9, which is significantly less than the random close packing of hard spheres where the CN is ~13–14. We note that the position of the 5th peak in the coordination shell remains unchanged over the entire temperature range. At low temperatures, e.g., 450 K, the 1st nearest neighbor and the 8th nearest neighbor are equally distant from the 5th nearest neighbor. Upon increasing temperature, the inner atoms move closer to the center atom in comparison with the outward displacement of the 8th nearest neighbor, exhibiting an obvious core-softening effect and a distinctly different temperature dependence from simple liquids like liquid Cu (Fig. 4b).

Core softening, which has a physical origin in the electronic structure, for instance, due to the Friedel oscillation[14], has been

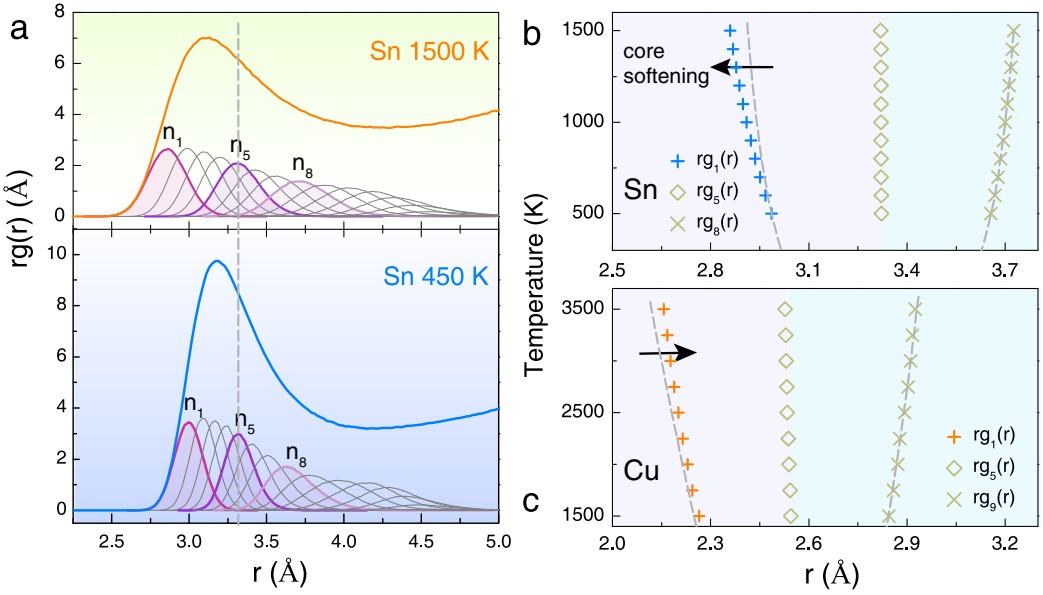

**Fig. 4 Core-softening effect of liquid Sn observed from isochoric AIMD simulation. a** Simulated $rg(r)$ of *l*-Sn (V = 29 Å³/atom) at 450 K and 1500 K, respectively. Also plotted are the Gaussian-like radial distribution functions of the *i*th nearest neighbor (denoted as $n_i$), $rg_i(r)$. The center position of the 5th nearest neighbor ($n_5$) is found to be constant at all temperatures, which is used to gauge the relative shift of the other nearest neighbors. **b**, **c** Average positions of the nearest neighbors of liquid Sn and Cu at different temperatures. The dashed lines show the harmonic effect (i.e., symmetric positions) of the nearest neighbors with respect to $n_5$ in the liquid. For *l*-Sn at low temperatures (e.g., 450 K), the 1st nearest neighbor and 8th nearest neighbor are equally distant from the center position ($n_5$). Upon increasing temperature, the inner nearest neighbors move more aptly toward the core, demonstrating a core-softening effect. This is in direct contrast with *l*-Cu, where the inner nearest neighbors feel more repulsive forces and move farther from the harmonic line at elevated temperatures.

commonly associated with anomalous liquids[28]. According to potential mean force theory[61], $g(r) = e^{-\beta u(r)}$, the observed core-softening phenomenon of liquids can be simulated by core-softened pair potentials. Indeed, in early simulations of the Sn-type liquid structure factors, a ledged potential was introduced[7]. Later on, other variations of core-softened potentials were proposed to cope with the complexities of anomalous liquids[7,16,28]. Our experimental and simulation results not only confirm the core-softening effect in *l*-Sn, but also lay the groundwork for the more detailed 3D structural analysis that follows.

**A folded-network structure model for liquid Sn**. To elucidate the core-softening phenomenon and expound its implications on the liquid structure, we sought to understand why short bonds are preferred at high temperatures in *l*-Sn. The chemical bonding in *l*-Sn was rigorously analyzed with quantum mechanics, based on electron localization function (ELF) analysis (see Supplementary Note 6). Enhanced electron localization in the middle of two atoms indicates that atomic bonding tends to be more covalent-like (see Supplementary Fig. 12). Plotting bond character ($\chi$) against interatomic distance (Fig. 5a), one can immediately see that the bonding character of *l*-Sn has a strong correlation with bond length, exhibiting a continuous distribution radially decaying with increasing $r$. Per the definition of ELF, we can see that strong covalent bonding is preferred when the interatomic distance is short, whereas metallic bonding is the norm at large distances. Setting $\chi = 0.6$ to delineate the covalent bonding from metallic bonding, the coordination shell can be separated into two bonding groups, and the covalently bonded atoms dwell in the inside of the coordination shell. More interestingly, as shown in the $\chi$–$r$ plot, contrary to the conventional wisdom that covalent bonding disappears at high temperatures, the covalent bonds not only persist at high temperatures but also greatly increase in

number, which exactly underlies the core-softening effect demonstrated in Fig. 4.

The covalent bonds are found to have a strong angular correlation, as seen from the bond-angle distribution functions (Fig. 5b). Such strong bond-angle correlations may not be adequately described by core-softened pair potentials but are uncompromisable for establishing the 3D structure of liquids[62]. For strong covalent bonds, we can see that the bond angles are centered around 109°, characteristic of tetrahedral bonding. This trait is reminiscent of the tetrahedral network described by the CRN model. Nonetheless, different from the CRN structure, the bond angles of the covalent bonds in liquid Sn show certain flexibility, largely due to the presence of non-bonding free electrons. The overall bond-angle distribution, $g_{local}^{(3)}(\theta)$, has a bimodal distribution profile (Supplementary Fig. 13), with two peaks centered at 100° and 60°, respectively, reflecting the flexible nature of the covalent bonding and metallic bonding.

Given the bonding behavior—that is, the bonding character changes radially from covalent to metallic and angularly follows the prescribed bond-angle distribution—a new atomic picture emerges, denoted a folded-network (FN) structure (illustrated in Fig. 6), lying in the middle-ground of the HS model and the CRN model. For comparison, we provide a CRN model of low-density amorphous (LDA) Sn, which has nearly perfect tetrahedral coordination (see Supplementary Note 7 for details about obtaining the CRN model of LDA-Sn). Networks can be characterized by the ring statistics, for instance, following King's definition of rings[63]. In the CRN model of LDA-Sn, five-, six-, seven-member rings are dominant, whereas, such rings do not exist in HS-like atomic packing. Likewise, the identity of the folded network can be revealed by ring statistics in which the smaller-sized four-, five-, six-member rings are shown to be a prominent structural feature (Supplementary Table 1).

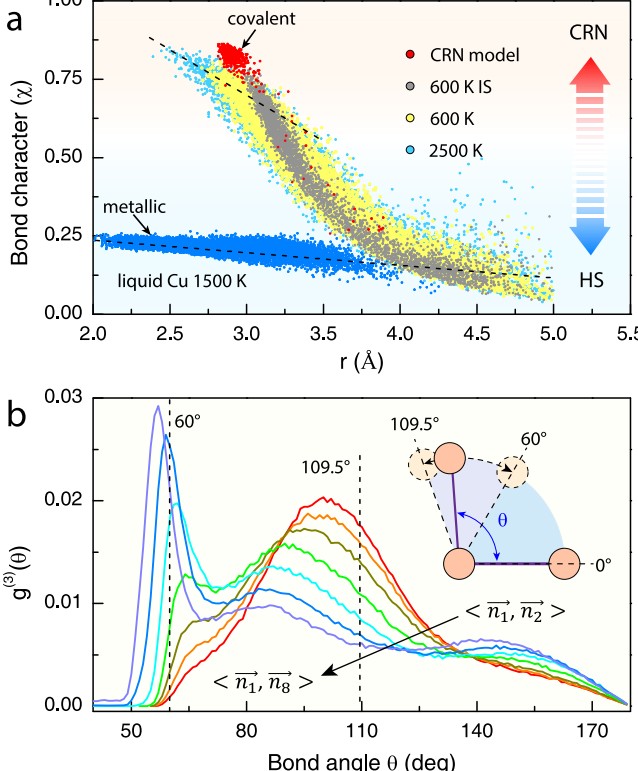

**Fig. 5 Bonding characteristics of liquid Sn. a** Bond character ($\chi$) as a function of distance ($r$). Also shown are the covalent bond character for the CRN model of *l*-Sn and the metallic bond character for *l*-Cu with the HS model. The $\chi$ in *l*-Sn falls in-between the CRN and HS models due to mixed covalent and metallic bond characters. The bonding behavior is found to be common to other polyvalent liquid metals (see Supplementary Fig. 18). **b** Bond-angle distribution function $g^{(3)}(\theta)$ of *l*-Sn at 600 K. $\langle \vec{n_1}, \vec{n_j} \rangle$ is the angle formed between the shortest covalent bond and the bond with nearest neighbor $n_j$ (from red to purple). The $g^{(3)}(\theta)$ of the HS model centers around 60°, and that of the CRN model centers around the tetrahedral bond angle at 109.5°. Atomic bond angles in *l*-Sn show large flexibility, exhibiting a jackknife-like foldability as illustrated in the inset.

The RDFs of *l*-Sn agree with the folded-network model. To better resolve the structural characteristics of *l*-Sn, we obtained the inherent structures (IS) of the liquid (Supplementary Fig. 14), which were found to be insensitive to the temperature at which the liquid was studied. The peak positions on the $g(r)$ of the IS exhibit characteristic ratios of 1:1.5:2, a manifestation of the tetrahedral arrangement of the covalent bonds and other longer metallic bonds opposing them in the folded network, as illustrated in Supplementary Fig. 13. The structural distinctions among the FN, CRN, and HS models are also evidenced from the comparisons of the RDFs and $g^{(3)}_{local}(\theta)$.

We now proceed to address an important question concerning the liquid ergodicity of the folded-network model of Sn—that is, how does the liquid maintain the folded-network structural identity while attaining its fluidity? The key lies in the fluctuating bonding nature of the liquid. The relaxation dynamics and lifetimes of the atomic bonds (Supplementary Fig. 15) have been revealed by analyzing the bond auto-correlation function (see Supplementary Note 8). During short times, we find that short covalent bonds relax slightly slower than the metallic bonds; but for longer relaxation times (e.g., 1.0 ns at 600 K), the relaxation dynamics of both covalent and metallic bonds converge, primarily owing to the mutual transitions between covalent and metallic

characters of the same bonds occurring in an ephemeral picosecond timescale (less than a picosecond at 600 K). This bond fluctuation behavior can also be understood in terms of ballistic cage-rattling of the atoms. As such, the atomic bonds in the liquid exhibit a fluctuating covalent/metallic character, which gives rise to its fluidity (Supplementary Fig. 16). Such a finding bears resemblance to water, where bifurcated hydrogen bonds have been well-known[64–66], suggesting a common mechanism for the relaxation of network-forming liquids.

Having presented the atomic structure of *l*-Sn, we are able to shed light on the property changes as well as the phase transition in the context of a folded network. The peak split observed in the second peak of the $g(r)$ (as shown in Fig. 1) is attributed to enhanced covalent bonding in the liquid. At high temperatures, owing to an increased number of short bonds (Fig. 5), the separation of the two subgroups in the coordination shell becomes more pronounced, resulting in the peak split observed in the experiment. By setting $\chi = 0.65$, we singled out the strongest bonds in the liquid. Counterintuitively, the number of these covalent bonds is found to increase, with rising temperature. As the fraction of the covalent bonds reaches a threshold (taken as 0.18 in this work), a percolating covalent bond network is developed (Supplementary Fig. 17). The percolation of the covalent bonds thus signals a percolation type transition known for its robustness for a continuous phase transition toward a spanning phase underlying the LLT of *l*-Sn[67,68].

The folded-network model of liquids has its origin in the electronic structure as illustrated in Fig. 5. Interestingly, this bonding behavior was found to be general to most post-transition polyvalent elements, where the bond character as a function of bond distance falls on a master curve (Supplementary Fig. 18). While the physical explanation for this universal bonding behavior calls for in-depth investigations, the liquid structures of these systems can be generally described by the folded-network model. Such an atomic model can be readily extended to even more complex systems containing covalent bonding, such as liquids and glasses of chalcogenides, sulfides, or biopolymers[23].

In terms of phase behavior, the presented folded-network atomic picture appears to reinforce the concept of patchy particles for tetrahedral liquids recently proposed by Smallenburg et al.[33], where rich phase behavior was predicted, depending on the angular flexibility. Hence, the newly proposed structural model holds the key to understanding polyamorphism in terms of atomistic mechanisms. For instance, the bonding unfolding/ folding mechanism can be directly applied to polyamorphic phase transitions, such as the LDA-HDA (high-density amorphous) transitions observed in Si[34,69], Ge[70], and Sn (Supplementary Fig. 19) where the bond-folding process assumes the short atomic range, but may also be applicable to disordered systems with folding processes taking place in the intermediate structural range ($r > 3$ Å), such as protein folding, where structural fluctuations occur among tiny patches with distinct bonding characteristics[23,71].

## Methods

**In situ temperature-resolved synchrotron X-ray diffraction**. The structure evolution of liquid Sn was monitored by in situ angle-dispersive X-ray diffraction at high temperatures at the 11-ID-C beamline with an X-ray energy of 110 keV at the Advanced Photon Source. The initial sample was a tin wire (99.9985%, Alfa Aesar) 0.4 mm in diameter and 6 mm in length. The samples were sealed in a vacuum quartz capillary (0.5 mm in outer diameter and 0.01 mm in wall thickness, Hilgenberg GMBH) and placed onto a TS 1500 heating stage (Linkam Scientific Instruments). In situ synchrotron XRD ($\lambda = 0.1174$ Å) patterns were recorded from room temperature to 1320 K at a rate of 8 K/min. The beam size was about $0.2 \times 0.2$ mm. Diffraction patterns were collected using a 2D detector with a sample-to-detector distance of 492.4730 mm. X-ray scattering data were obtained with a total of 10 s exposure time (1 s/frame, 10 frames/file and 1 file/dataset). All raw data were integrated and converted to intensity $I(q)$ versus scattering wave

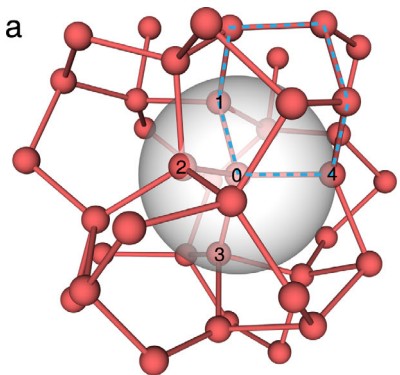 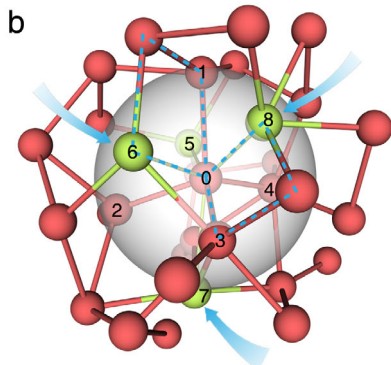

**Fig. 6 Folded-network structure model for molten Sn. a** A continuous random network (CRN) structure constructed for LDA-Sn. The semi-transparent sphere shows an imaginary coordination shell where four coordination atoms are located (denoted by the digits 1–4). A six-member ring characteristic of the CRN model is traced by the blue dashed lines. **b** The folded-network structure for high-temperature *l*-Sn. The coordinate number is 8–9, and four-member rings (traced by the dashed lines) are a noticeable structural feature of the folded network. The arrows indicate that the atoms/bonds fold back toward the coordination shell, forming direct metallic bonding with the center atom. The folded-network structure is derived as a logical consequence of the particular bonding behavior of liquids as shown in Fig. 5 and Supplementary Fig. 18.

vector q using the Fit2D software, and the experimental structure factors $S(q)$ were derived employing an in-house computational code. The uncertainty of the temperature for the XRD measurement was about ±3 K.

**Temperature-resolved small-angle X-ray scattering.** Small-angle X-ray scattering measurements were carried out at 12-ID-B at the APS. Beam energies and sample-to-detector distances of 13.3 keV and 2033 mm were used. Sample-to-detector distances were calibrated using a silver behenate standard. A tin wire sealed in a vacuum quartz capillary (0.1 mm in outer diameter and 0.01 mm in wall thickness) was mounted on a Linkam TS 1500 heating stage. SAXS patterns were collected from 530 to 1000 K at a heating rate of 4 K/min. To reduce the possibility of radiation damage, the data were continuously collected for five frames and all scattering images were averaged afterward. The scattering of the empty capillary at each temperature was measured separately and subtracted. All raw data were integrated using an in-house code based on the MATLAB software. The transmission factor of the samples was determined using an ion chamber located in front and behind of the sample.

**Heat capacity ($C_P$) measurement.** The $C_P$ at high temperature was acquired using Setaram 96 line evo under a helium atmosphere at a flow rate of 20 mL/min. The sample mass was about 2400 mg with 5 mm in diameter and 17 mm in length. The sample was placed in $Al_2O_3$-lined Pt crucibles and the temperature was raised at a constant heating rate of 4 K/min from RT to 1000 °C. Signals of the empty crucibles taken prior to the experiments were subtracted from the measurements to obtain a straight baseline. The differential scanning calorimetry (DSC) was calibrated with a sapphire standard under the same condition. The excellent symmetry of heat-flow detectors, each composed of three thermocouples, guarantees very good baseline stability and accuracy for thermal measurement. The specific heat measurements were conducted at least three times with different samples and settings, and the results were reproducible.

In addition to the continuous scan mode stated above, in order to ascertain the divergence behavior of $C_P$, a lengthy stepwise-scanning mode[72] was adopted using Setaram 96 line evo with large sample masses. This approach was similar to the specific heat measurement of liquid sulfur for quantifying a second-order phase transition[73]. Key to this method is the thermal equilibration of the sample at each temperature. Because of large-volume samples used for signal enhancement, this procedure requires very long equilibration time. The temperature was first raised from RT to 730 K at a rate of 10 K/min and kept at 730 K for 2 h to reach full thermal equilibrium. Then, heat flow signals were acquired with a stabilization time period of 1.0 h at each temperature. After each step, the sample was slowly heated to the next temperature with a 5 K interval at a heating rate of 1 K/min up to 830 K. To complete the process, the same protocol was repeated for both the calibrant and the blank cell. Upon integration of the heat flow curve, the total heat Q consumed in the interval was obtained. Dividing Q by the temperature interval $\Delta T$ and the mass of the specimen gives the specific heat. Detailed data analysis can be found in Supplemental Fig. 7.

**In situ sound velocity measurements at high pressure and high temperature.** A lump of tin cut from a tin plate of 99.9985% purity (Alfa Aesar) was used as the starting material for sound velocity measurements. The initial length of the sample in the sound wave propagation direction was 1.5 mm and the length at HPHT conditions was determined by the EOS of tungsten carbide (WC)[74]. No distortions or changes of the WC slot were observed after the experiments, and the depth of the WC slot remained at1.5 mm. More details can be found in ref. [75].

**Ab initio molecular dynamics simulation.** Ab initio molecular dynamics simulations were conducted with the density functional theory (DFT) based Vienna Ab Initio Simulation Package (VASP)[76]. We used the Perdew-Wang type exchange-correlation functional[77] for the generalized gradient correction and the projector-augmented waves (PAW) potential[78,79] for Sn. The valence electrons of Sn were $5s^2 5p^2$. The simulated system typically contained 288 atoms, and only the Γ-point was sampled for the liquid. For NPH (constant pressure and enthalpy) simulations, we used a large energy cutoff to include sufficient plane waves to overcome the Pulay stress effect.

## Data availability

All relevant data are available from the authors, and/or are included within the manuscript and Supplementary Information.

## Code availability

The software used for data analysis is available from H.S. upon request.

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

## Acknowledgements

This work was supported by the Foundation of National Key Laboratory of Shock Wave and Detonation Physics (Grant Nos. 6142A03180101 and JCKYS2018212002), and the Science Challenge Project (TZ2018001). W.Y. was supported by the NSFC under Grant No. 51772184. H.S. was supported by the NSF under Grant No. DMR-1611064. L.X. thanks X.H. Li, Y. Wang, L. Liu, X. Chao, C. Meng, W. Zhu, and X.Y. Li for their assistance and discussions, M. Hou, Q. Hu, H. Dong, F. Han, and F. Guo for technical support in the XRD experiment, Y. Tao, Y. Chen, H. Lou, and Q. Zeng in the DSC experiment. XRD measurements were performed at Advanced Photon Source (APS), Argonne National Laboratory. Specific heat measurements were performed at Shanghai Jiaotong University and Shanghai Institute of Ceramics, Chinese Academy of Sciences.

## Author contributions

Q.W. and H.S. conceived and designed the project and directed the experiments. L.X. and Z.W. carried out the high-pressure sound velocity measurements experiment. L.X., W.Y., Y.R., and J.Z. conducted high-temperature XRD measurements. J.C, H.S., J.Z., and X.Z. conducted SAXS experiments. L.X. and S.C. carried out heat capacity measurements. H.S. performed the computer simulation. L.X. and H.S. performed the experimental data analysis. L.X. and H.S wrote the manuscript. All authors contributed to the discussion of the results and revision of the manuscript.

## Competing interests

The authors declare no competing interests.
