## [Peer Review File · Nature Communications]

Folded network and structural transition in molten tinEditorial Note: This manuscript has been previously reviewed at another journal that is not operating a transparent peer review scheme. This document only contains reviewer comments and rebuttal letters for versions considered at *Nature Communications*.

REVIEWERS' COMMENTS

Reviewer #1 (Remarks to the Author):

I believe the authors have done a good job in softening their claims, switching the focus from a unproved liquid-liquid critical point to a properly demonstrated structural cross-over. I have read the responses to all reports and in my opinion I have found satisfactory answers to all of them. The results, especially the experimental ones, are relevant and will motivate further investigation on this sample. I believe the present version of the manuscript is well suited for publication in its present form in *Nature Communications*.

Reviewer #2 (Remarks to the Author):

In the new version of the manuscript, the authors have retracted from the claim of a second-order LLPT. However, it is now called a "second-order-like LLT". As evidence for this, tiny features in the static structure factor/radial distribution function as well as in the specific heat and the isothermal compressibility are shown. I agree that these features may indicate gradual structural changes, but I do not see any evidence for any avoided critical phenomenon or even a phase transition, e.g. there is no sign for any growing correlation length. In my previous report, I have mentioned the simulation study by Sukhomlinov and Muser who reported similar phenomena for a metallic system, especially a similar small anomaly in the specific heat. It would have been interesting to discuss this work in the manuscript and compare their results to the present findings on molten tin.

In conclusion, I think the authors do not provide a correct interpretation of their experimental data and therefore, I cannot recommend the manuscript for publication.

Response to the second reviewer's comments

In the new version of the manuscript, the authors have retracted from the claim of a second-order LLPT. However, it is now called a "second-order-like LLT". As evidence for this, tiny features in the static structure factor/radial distribution function as well as in the specific heat and the isothermal compressibility are shown. I agree that these features may indicate gradual structural changes, but I do not see any evidence for any avoided critical phenomenon or even a phase transition, e.g. there is no sign for any growing correlation length. In my previous report, I have mentioned the simulation study by Sukhomlinov and Muser who reported similar phenomena for a metallic system, especially a similar small anomaly in the specific heat. It would have been interesting to discuss this work in the manuscript and compare their results to the present findings on molten tin.

In conclusion, I think the authors do not provide a correct interpretation of their experimental data and therefore, I cannot recommend the manuscript for publication.

Reply:

We appreciate the comments made by the reviewer. In his/her view, the structural change, as evidenced from structure factors, density inflections, as well as thermodynamic response functions obtained from both experiment and simulation, can be explained by a structural cross-over. As stated in our previous replies to all the reviewers, we have actually softened our tone and stepped away from our original claim that the transition belongs to a thermodynamic second-order phase transition. The structural change is now loosely referred to as a structural transition which endows the meaning of a structural cross-over. The rigorous interpretation of the nature of the transition (i.e., whether it can be truly regarded as a second-order transition) awaits future in-depth investigations. As the other reviewer pointed out, in the revised manuscript, we have switched the focus from an unproved liquid-liquid critical point to a properly demonstrated structural cross-over. In this regard, our interpretation is indeed in line with the reviewer's viewpoint. Additionally, we would also like to remind the reviewer that we have proposed a new structural model based on the counterintuitive bonding nature of the element, which is the origin for the anomalies observed in the liquid.

Since we no longer insist on a second-order phase transition, the growing correlation length is not a major concern anymore. We agree with the reviewer that the growing correlation length problem has been extremely difficult to study. For instance, the existence of a growing correlation length within the context of glass-transition has been actively investigated and hotly debated. We would like to leave this problem for future studies.

We have studied the paper by Sukhomlinov and Muser, titled “Anomalous system-size dependence of properties at the fragile-to-strong transition in a bulk-metallic-glass forming”. The paper deals with the nature of a fragile-to-strong transition in the Zr-Cu-Al metallic liquid. It should be noted that the paper used the model potential developed by one of the authors (H. Sheng) of this work. The specific heat jump corresponds to the Arrhenius-to-non-Arrhenius crossover when the Zr-Cu-Al liquid enters the supercooled regime (i.e., the liquid falls off equilibrium). We believe that, while the results reported in molten tin are phenomenologically similar to the fragile-to-strong (FTS) cross over in Zr-Cu-Al, the structural origins are quite different. In this work, we have identified that the specific heat anomaly originates from a covalent-metallic bond percolation transition, whereas in the FTS of supercooled liquid, the structural origin remains unclear. We thus refrain from further complicating the phenomena we observed in molten tin and prefer not to make connections with other unaddressed physics such as glass transition and FTS cross-over.